# Improving measurement of functional status among older adults in primary care: A pilot study

Rebecca T. Brown[ID]1,2,3,4*, Kara Zamora5,6☉, Anael Rizzo5,6☉, Malena J. Spar5,6, Kathy Z. Fung5,6, Lea Santiago5, Annie Campbell7, Francesca M. Nicosia5,8

**1** Geriatrics and Extended Care Program, Corporal Michael J. Crescenz VA Medical Center, Philadelphia, Pennsylvania, United States of America, **2** Center for Health Equity Research and Promotion, Corporal Michael J. Crescenz VA Medical Center, Philadelphia, Pennsylvania, United States of America, **3** Division of Geriatric Medicine, Perelman School of Medicine of the University of Pennsylvania, Philadelphia, Pennsylvania, United States of America, **4** Leonard Davis Institute for Health Economics, University of Pennsylvania, Philadelphia, Pennsylvania, United States of America, **5** San Francisco VA Health Care System, San Francisco, California, United States of America, **6** Division of Geriatrics, University of California, San Francisco, San Francisco, California, United States of America, **7** Martinez VA Medical Center, Martinez, California, United States of America, **8** Institute for Health & Aging, School of Nursing, University of California, San Francisco, California, United States of America

☉ These authors contributed equally to this work.
* rebecca.brown@pennmedicine.upenn.edu

**Data Availability Statement:** "The underlying data for this study consists of (1) in-depth, qualitative interviews with (a) Veterans with aging-related functional impairment and their caregivers and (b) employees of the U.S. Veterans Health

## Abstract

Despite its importance for clinical care and outcomes among older adults, functional status–the ability to perform basic activities of daily living (ADLs) and instrumental ADLs (IADLs)–is seldom routinely measured in primary care settings. The objective of this study was to pilot test a person-centered, interprofessional intervention to improve identification and management of functional impairment among older adults in Veterans Affairs (VA) primary care practices. The four-component intervention included (1) an interprofessional educational session; (2) routine, standardized functional status measurement among patients aged ≥75; (3) annual screening by nurses using a standardized instrument and follow-up assessment by primary care providers; and (4) electronic tools and templates to facilitate increased identification and improved management of functional impairment. Surveys, semi-structured interviews, and electronic health record data were used to measure implementation outcomes (appropriateness, acceptability and satisfaction, feasibility, fidelity, adoption/reach, sustainability). We analyzed qualitative interviews using rapid qualitative analysis. During the study period, all 959 eligible patients were screened (100% reach), of whom 7.3% (n = 58) reported difficulty or needing help with ≥1 ADL and 11.8% (n = 113) reported difficulty or needing help with ≥1 IADL. In a chart review among a subset of 50 patients with functional impairment, 78% percent of clinician notes for the visit when screening was completed had content related to function, and 48% of patients had referrals ordered to address impairments (e.g., physical therapy) within 1 week. Clinicians highly rated the quality of the educational session and reported increased ability to measure and communicate about function. Clinicians and patients reported that the intervention was appropriate, acceptable, and feasible to complete, even during the COVID pandemic. These findings suggest that this

Administration; and (2) quantitative measures of functional status among Veterans. For both the qualitative and quantitative data, it is not possible to create a minimal data set as this study did not obtain ethical approval or informed consent from participants to publicly share underlying data sets. For the qualitative data, relevant excerpts from transcripts are included within the paper. The qualitative and quantitative datasets generated and/or analyzed during this study are not publicly available but may be available upon request at the Center for Health Equity Research and Promotion of the U.S. Department of Veterans Affairs administrative offices, at 215-823-5817 (https://www.cherp.research.va.gov/)."

**Funding:** This research was supported by the Veterans Affairs (VA) Quality Enhancement Research Initiative (grant number QUE 15-283 to RTB; https://www.queri.research.va.gov/). The funding source had no role in the study design, data collection and analysis, decision to publish, or preparation of the manuscript.

**Competing interests:** The authors have declared that no competing interests exist.

intervention is a promising approach to improve identification and management of functional impairment for older patients in primary care. Broader implementation and evaluation of this intervention is currently underway.

## Introduction

Functional status–the ability to perform daily activities such as bathing, dressing, and preparing meals–is central to older adults' quality of life, health, and independence [1–3]. Older adults who develop functional impairment, defined as having difficulty or needing help performing these activities, are at higher risk for hospitalization, long-term institutionalization [4,5], and death [6,7]. Maintaining functional status is one of the health outcomes that matters most to older adults [1] and plays a key role in their decision-making about treatments and advance care planning [2,3]. Proactively identifying functional impairment allows clinicians to deliver interventions, such as physical and occupational therapy, to help delay or prevent nursing home admission [8–10]. Understanding functional status is also essential to delivering person-centered care, including evaluating how patients will tolerate interventions [11–15], individualizing cancer screening and treatment [16,17], assessing prognosis [6,7,18], understanding frailty, and determining the need for long-term services and supports [19]. On a population level, health systems can use information about function to anticipate service needs [20]. Yet despite ongoing calls for standardized functional status measurement among older adults, uptake in the U.S. has been slow and inconsistent [20–22].

The Veterans Health Administration (VA), the largest integrated health system in the United States (US), is a leader in implementing and evaluating a range of routine screening and assessment efforts in outpatient settings, such as screening for alcohol misuse and housing instability [23–25]. In 2009, the VA Central Office of Geriatrics and Extended Care requested that primary care practices implement annual screening for functional status among older veterans in primary care using an electronic tool administered during patient triage [26]. The tool was based on the Katz activities of daily living (ADL) and Lawton and Brody instrumental ADL (IADL) screening instruments, formatted with checkboxes with response options corresponding to ability to perform each ADL (bathing, dressing, transferring, toileting, eating) and IADL (shopping, preparing food, managing medications, managing finances, doing housework, using transportation, using the telephone) [27,28]. While important, these initial efforts involved limited stakeholder input on design and implementation and did not formally assess barriers and facilitators prior to implementation. A national evaluation later showed that uptake of the screening tool was low and of varying quality [21,22,26]. This screening approach also had low sensitivity, identifying only 45% of older patients with ADL impairment [21].

As part of a VA study to improve measurement of functional status in VA primary care, we used qualitative methods to systematically examine barriers and facilitators to measuring functional status at sites that were and were not using the new screening tool. We identified provider- and system-level barriers to routine functional status measurement in primary care, including time pressures, cumbersome tools, and the perception that measurement would not be used to inform care [21,22]. We also identified patient preferences for measurement of functional status, including face-to-face screening, asking questions about both difficulty and need for help with daily activities, and providing context when asking potentially sensitive questions about function [29]. These findings informed the development of a multi-component intervention to improve measurement of functional status in VA primary care clinics

[30]. This intervention includes clinician education, annual functional status screening with a low-burden electronic questionnaire, follow-up assessment and electronic health record (EHR)-based referral menus when impairments are identified, and tailored reports for clinicians and operational leaders. In the current study, we conducted a pilot study to test this intervention's appropriateness, acceptability, feasibility, fidelity, adoption, and sustainability for patients and clinicians.

## Methods

### Design overview

We pilot-tested the intervention at two VA primary care practices in Northern California from July 30, 2020, to September 30, 2020. Because implementation of the intervention was part of a larger quality improvement initiative, it was exempt from IRB review in compliance with VA Handbook 1200.05 [31]. The evaluation of the intervention was considered research and received approval from the institutional review boards of the San Francisco VA Medical Center and the University of California, San Francisco (combined approval no. 15–17697) and the institutional review board of the Corporal Michael J. Crescenz VA Medical Center (approval no. 1581262–6). All patients and VA employees provided verbal consent prior to enrollment in the study. Human subjects' data were collected between July 1, 2020 and September 30, 2020. The authors had access to information that could identify individual participants during and after data collection.

### Recruitment and eligibility

Licensed vocational nurses (LVNs) and primary care providers (PCPs; i.e., physicians, nurse practitioners) were eligible if they worked at least one half-day per week at one of the two practices. Practice-wide implementation of the intervention (i.e., for all nurses and PCPs in each practice) was initially planned at both sites. However, due to the impact of the COVID-19 pandemic on clinic procedures and workloads, clinic leadership provided permission to invite a limited number of nurses and PCPs in each practice to pilot test the intervention. The duration of the pilot was also shortened from 6 months to 3 months due to COVID-related delays in the start date. In July 2020, clinicians were invited to participate via e-mail. Because implementation of the intervention was considered quality improvement, clinicians provided assent to participate. Among clinicians who assented, electronic tools and templates for the intervention were assigned to their EHR profile. In addition to nurses and PCPs, we invited social workers and clinic leaders (nursing and medical directors) to provide feedback on the intervention. All participants provided verbal consent to participate in the evaluation, which was witnessed and documented by research staff in an electronic database. Among patients, we assessed ability to provide informed consent using a teach-back method [32]. We did not assess ability to provide informed consent among staff members, since they are not considered a population at risk for lacking ability to provide informed consent. These consent procedures were approved by our IRB. Nurses and social workers received up to $100 for their participation ($25 for pre-implementation surveys, $25 for post-implementation surveys, $50 for qualitative interviews), and PCPs and clinic leaders received up to $225 ($50 each for pre- and post-implementation surveys; $125 for qualitative interviews).

Patients were eligible to receive the intervention if they were aged 75 and older. Patients who had a clinic appointment scheduled with their PCP (routine or urgent visit) during the trial period and had the electronic questionnaire administered by clinic staff were eligible to participate in an interview about their experience with the intervention. We sent mailings to patients' home addresses including a letter explaining the study with a toll-free opt-out

telephone number and study consent form. If patients did not opt-out within one week of the date the letter was sent, we called to invite the patient to participate. For patients interested in participating, we used a teach-back method to obtain informed consent [32]. Patients who were unable to provide informed consent had the opportunity to provide assent for a legally-designated proxy to participate in an interview on their behalf. Patients and proxies received $50 to complete the interview.

## Intervention

The development of the intervention and details of the intervention components have been previously described in detail [30]. The intervention was designed to incorporate patient and clinician preferences for functional status measurement and to address barriers to previous measurement approaches (Fig 1) [22,29]. The four components included: (1) an interprofessional educational session; (2) routine, standardized functional status measurement; (3) screening by nurses and follow-up PCP assessment when impairments were identified; and (4) electronic tools and templates to facilitate screening and assessment. Of note, the intervention was intended to include a fifth component, tailored reports for clinic and health system leaders, including information on clinician adoption of screening and patient functional status at the level of the clinic and medical center. However, it was not possible to create reports during the pandemic due to changes in workflow and competing demands. As an implementation strategy for the intervention, an LVN at each clinic served as a clinic-based champion. We chose LVNs for this role because they triage patients and administer clinical screening tools, and thus their role has the potential for the largest impact on workflow and screening practices. The champion collaborated with the research team to engage members of the primary care team and provide feedback on study implementation.

## Data collection and measures

Data collection included four components: (1) an evaluation of the educational session for LVNs and PCPs; (2) a pre-implementation and post-implementation survey of clinicians (i.e., clinic staff with patient contact, including LVNs, PCPs, and social workers) about their perspectives on the intervention; (3) a semi-structured exit interview among clinicians and a subset of patients; and (4) extraction of EHR and administrative data. Of note, the original data collection plan also included clinic-based ethnographic observations of clinician workflows, use of the electronic tool and referral menu by clinicians, and communication about function between patients and clinicians. Due to the COVID-19 pandemic, we were unable to conduct in-person observations and we adapted several measures as described below.

We assessed several types of measures for each data collection component. The educational session evaluation focused on its quality and effectiveness. We assessed implementation outcomes using the pre- and post-implementation surveys, semi-structured interviews, and EHR and administrative data. Implementation outcomes were based on Proctor's taxonomy [33] and included six measures: (1) appropriateness, defined as the perceived fit, relevance, or compatibility of an intervention for a given setting; (2) acceptability and satisfaction, defined as the perception by recipients that an intervention is agreeable or satisfactory; (3) feasibility, defined as the extent to which a new intervention can be carried out within a given setting; (4) adoption or uptake, defined as clinicians' intention or action of taking up an intervention; (5) fidelity, defined as the degree to which an intervention is implemented as intended by the program developers; and (6) sustainability, defined as the extent to which a newly-implemented intervention is maintained within ongoing operations [33].

| **1. Routine, standardized functional status measurement** |
| --- |
| • **Component 1:** Annual measurement with a standardized tool<br>• **Rationale:** Addresses barriers to screening and assessment (competing priorities, a lack of standardized processes) and to documenting and using data (lack of standardized data location, poor team coordination); contributes to "meaningful measurement" by allowing use of standardized data to improve care |
| **2. Licensed vocational nurse (LVN) screening and follow-up PCP assessment** |
| • **Component 2:** Annual LVN screening during patient triage and follow-up PCP assessment and referral(s)<br>• **Rationale:** Team-based approach clarifies team roles and responsibilities and fosters interprofessional environment |
| **3. Electronic tools and templates to facilitate LVN screening, PCP assessment, and documentation** |
| • **Component 3a:** <u>LVN screening tool:</u> (1) validated brief two-question pre-screener asking about difficulty performing ADLs and IADLs,[68] and, among patients who report difficulty on pre-screener, (2) in-depth screener asking about difficulty and needing help with each ADL/IADL. Results used to auto-populate nursing note.<br>• **Rationale:** Two-part screener intended to quickly identify patients with impairment who would benefit from in-depth screening while "screening out" individuals without impairment, reducing LVN screening burden |
| • **Component 3b:** <u>PCP alert and referral menu:</u> If patient screens positive (i.e., reports difficulty/needing help with ≥1 ADL/IADL), PCP receives electronic alert linked to referral menu. Alert prompts PCP to review LVN screening results and perform additional assessment as needed. PCP can select up to 4 referrals: physical therapy, occupational therapy, social work, geriatric medicine.<br>• **Rationale:** Addresses stakeholder requests for integration of functional assessment into existing workflows and need for a standardized section to retrieve data on function; alert supports interprofessional approach to measurement; integrated EHR referrals address concerns that data will not be used to inform care and that clinicians lack knowledge of resources, making desired outcome (appropriate referral) more salient for PCPs |
| **4. Interprofessional educational session** |
| • **Component 4:** Brief pre-recorded educational session administered at beginning of implementation<br>• **Rationale:** Session reviews evidence for importance of measuring function, introduces core intervention components including the electronic tools and templates, and reviews patient perspectives on functional status measurement and role of interprofessional communication |
| **5. Tailored reports (note: unable to develop due to COVID pandemic)** |
| • **Component 5:** Automated reports pulled at level of medical center, clinic, and/or individual clinician to report varying statistics (e.g., proportion of Veterans needing help with ADLs)<br>• **Rationale:** Reports provide access to population-level data to inform strategic planning and efforts to keep patients functional at home rather than in more costly institutional care |

**Fig 1. Components of an intervention to improve measurement of functional status in VA primary care and rationale.**

**Educational session evaluation.** The educational session evaluation included closed-ended and open-ended questions, which LVNs and PCPs completed via secure Research Electronic Data Capture (REDCap) survey links after completing the educational session. The closed-ended questions included four domains, each rated on a 5-point Likert scale: participant demographics and professional experience; session quality (5 excellent, 1 poor); session effectiveness (5 very high, 1 very low); and retrospective pre-post evaluations of the

participants' knowledge and ability (5 definitely can, 1 definitely cannot). In the retrospective pre-post evaluations, participants were asked to compare what they knew before versus after the session. Studies show that compared to traditional pre-post evaluations, retrospective evaluations have better criterion validity and sensitivity to change [34].

In open-ended items, participants described their attitudes about the effectiveness of the intervention ("do you think that the new approach will improve assessment of function for older adults in your clinic") and changes in intended behavior ("list any changes you plan to make to your professional practice as a result of this session"). Studies show that attitudes and behavioral intention are highly correlated with subsequent behavior [35]. We also asked participants to comment on how the educational session and implementation process could be improved.

**Pre- and post-implementation surveys.** Clinicians received e-mails inviting them to participate in a brief pre- and post-implementation survey. The e-mail included a link to a secure web-based REDCAP survey instrument. Surveys included closed-ended questions focused on the intervention's appropriateness, acceptability and satisfaction, and feasibility, measured on a 5-point Likert scale (5, strongly agree, to 1, strongly disagree); we measured satisfaction on a 5-point scale (5, very satisfied, to 1, very dissatisfied). We developed questions based on domains from the Consolidated Framework for Implementation Research (e.g., intervention characteristics, inner setting, outer setting; see Table 4) [36].

**Semi-structured interviews.** During the last two weeks of the pilot study, clinicians received e-mails inviting them to participate in an interview to discuss their experience with the intervention. KZ and AR conducted semi-structured telephone interviews with interested clinicians. Interviews lasted approximately 30 minutes and were audio recorded. We first asked open-ended questions about the participants' professional background and general impressions of the intervention. We also asked questions to assess implementation outcomes including appropriateness; acceptability and satisfaction; feasibility, including adaptability of workflows and communication; fidelity of implementation; adoption, including ways in which functional status data were used to inform patient care; and sustainability of the intervention. We planned to assess fidelity via ethnographic observation, but this was not possible due to pandemic-related restrictions on in-person visits.

We also conducted semi-structured interviews to assess patients' and caregivers' experience with the intervention. We used purposive sampling to stratify recruitment by functional status [37], using existing data collected with the previous screening tool (i.e., independent, needs help with 1–2 ADLs, needs help with 3 or more ADLs). KZ, AR, and FMN conducted telephone interviews lasting 10–25 minutes. All interviews were audio recorded. We first asked open-ended questions about the patients' or caregivers' experience with functional status measurement, including whether they remembered being asked about ability to perform ADLs and IADLs, their opinion of being asked these questions, their impression of whether the nurse screening questions informed their interaction with the PCP, and how satisfied they were with their overall experience. We also asked closed-ended questions about demographics and functional status, using the same two-part tool used during the clinic visit to allow us to evaluate agreement between the clinic-collected and research-collected data. Unlike in the clinic, in the interview patients were asked about difficulty and need for help with all 6 ADLs and 7 IADLs, regardless of their response to the 2-item pre-screener.

**Measures from EHR and administrative data.** To measure adoption and fidelity, we used data extracted from the EHR and the VA's Corporate Data Warehouse. Nurse adoption of the screening tool was defined as the proportion of LVNs who completed screening of one or more eligible patients during the study period. We also assessed the proportion of eligible patients with functional status screening completed by an LVN (i.e., reach). PCP adoption was

defined as the proportion of PCPs who placed one or more referrals to address functional impairment among patients with positive screens for impairment. Because not all referrals entered in the VA EHR are captured in the Corporate Data Warehouse, we identified referrals by performing chart review among a subset of 50 patients who screened positive for functional impairment. In this subset, we identified any referrals to address functional impairments placed by the PCP within 1 week of the visit when the functional status questionnaire was administered and calculated the proportion of patients with functional impairment who received referrals. We acknowledge that management of functional impairment is complex and there is no "gold standard" for appropriate consults. For example, a patient may not require a consult because they are already receiving services. Functional status assessment may also inform clinical decision-making (e.g., whether to order cancer screening) but not result in a consult. Thus, to examine broader use of functional status data in clinical decision making, we also reviewed clinician notes (nurses and PCPs) in the subset of 50 patients for the visit when the questionnaire was administered to identify any references to functional status, ADLs, or IADLs.

Fidelity is defined as the degree to which an intervention is implemented as intended and has several dimensions including adherence and quality of delivery [33]. Because we intended the intervention to accurately capture functional status data that could be used to inform patient care, we assessed the accuracy of documented functional status as a dimension of fidelity. To do so, in the subset of patients who completed an exit interview, we assessed the agreement of the functional status measures collected during the clinic visit compared to a reference standard of the same measures collected during the patient's exit interview.

## Analysis

We used descriptive statistics to analyze participant characteristics, survey responses, and data extracted from chart review [38]. To evaluate the agreement of the clinic-collected and research-collected data, we used two approaches. First, we considered the research-collected data a reference standard and compared the sensitivity and specificity of the VA data to this standard. Second, we used kappa statistics to evaluate the agreement between the VA and research-collected data. Kappa statistics measure the agreement between separate ratings of the same construct beyond the agreement that would be expected by chance without designating one construct as the correct value. We conducted analyses using SAS 9.4 (SAS Institute, Inc., Cary, NC), Stata 12 (Stata Corp., Chicago, IL), and R 3.1.2 (R Foundation for Statistical Computing, Vienna, Austria).

We used rapid qualitative analysis methods developed for health services research to analyze qualitative interviews [39–41]. This is a team-based approach which allows qualitative results to be analyzed concurrently with data collection to inform the development and testing of interventions and implementation strategies. We organized interview data into summary templates, where a member of the project team listened to each audio file and summarized key points in the appropriate section of the template. To maintain rigor and trustworthiness, a second member of the team listened to the same audio file and reviewed the primary analyst's summary for accuracy. All team members reviewed each summary and met regularly to discuss results. To synthesize summary template data, we used matrix analysis, a method of displaying data to identify relationships, including commonalities and differences [42]. We organized matrices by stakeholder group and clinic to compare implementation outcomes across these groupings. This process, known as data reduction, is used to organize and highlight material so conclusions can be drawn from the data [43]. We paired findings with direct quotations to ensure alignment with participants' voices.

## Results

### Participant characteristics

Twenty-two clinicians participated in the pilot across the two medical centers, including nine LVNs, nine physicians, two nurse practitioners, one social worker, and one clinic leader (Table 1). More than three-quarters were women (77%), 50% were white, 28% Asian, 17% Native Hawaiian or Pacific Islander, and 6% Latino. Most had been in their current VA role for 5 years or fewer (61%), with 17% for 6–10 years and 28% for more than 10 years.

Nine-hundred fifty-nine patients were screened for functional impairment over the study period (Table 2). Among the 959 patients who were screened, mean age was 81.8 years (SD, 5.7), 91% were male, 74% white, 12% Black, 4% Asian or Pacific Islander, 0.5% Latino, and 9% other or unknown. The most common chronic conditions were diabetes (25%), coronary heart disease (25%), and chronic lung disease (18%). Based on the screening tool, 93% were independent in all ADLs, 7% had difficulty with 1 or more ADLs, 1% needed help with 1–2 ADLs, and 5% needed help with 3 or more ADLs. Eighty-six percent were independent in all IADLs, 14% had difficulty with 1 or more IADLs, 1% needed help with 1 or more IADLs, and 14% needed help with 3 or more IADLs.

We interviewed a subsample of 21 patients and/or caregivers (17 patients, 2 patient/caregiver dyads, 2 caregivers). The characteristics of the 21 patients who completed interviews were generally similar to those of the overall cohort in terms of age, gender, and race/ethnicity, but they had a higher prevalence of most chronic medical conditions (e.g., cerebrovascular accident, chronic pulmonary disease, arthritis) and a higher prevalence of functional impairment.

**Table 1. Clinician characteristics.**

| | Participants (n = 22)[a,b] |
|---|---|
| Role | |
| Licensed vocational nurse | 9 (41%) |
| Attending physician (MD/DO) | 9 (41%) |
| Nurse practitioner | 2 (9%) |
| Social worker | 1 (5%) |
| Clinic leadership | 1 (5%) |
| Woman | 17 (77%) |
| Race/ethnicity, n (%) | |
| White | 9 (50%) |
| Asian | 5 (28%) |
| Native Hawaiian or Pacific Islander | 3 (17%) |
| Latino/Latina | 1 (6%) |
| Years in current role | |
| ≤5 | 11 (61%) |
| 6–10 | 2 (11%) |
| >10 | 5 (28%) |
| Years' experience in field | |
| ≤5 | 5 (28%) |
| 6–10 | 3 (17%) |
| >10 | 10 (56%) |

[a]Percentages may not add to 100% due to rounding.

[b]Data are missing for 4 participants for race/ethnicity, years in current role, and years' experience in field.

**Table 2. Patient characteristics.**

| Characteristics | All patients (n = 959) | Interviewed patients (n = 21) |
|---|---|---|
| *Patients* | | |
| Age, mean years (standard deviation) | 81.8 (5.7) | 81.9 (6.1) |
| Woman, n (%) | 9 (0.9) | 0 (0) |
| Race/ethnicity, n (%) | | |
| White | 713 (74.4) | 16 (76) |
| Black | 118 (12.3) | 2 (10) |
| Latinx | 5 (0.5) | 1 (5) |
| Asian/Pacific Islander | 40 (4.17) | 1 (5) |
| Other/unknown | 83 (8.7) | 1 (5) |
| Marital status | | |
| Single or never married | 173 (18.0) | 1 (5) |
| Married or partnered | 416 (43.4) | 12 (57) |
| Widowed | 94 (9.8) | 2 (10) |
| Divorced | 270 (28.2) | 5 (24) |
| Missing | 6 (0.6) | 1 (5) |
| Chronic medical conditions | | |
| Coronary heart disease | 240 (25.0) | 2 (9.5) |
| Cerebrovascular accident | 146 (15.2) | 4 (19.1) |
| Diabetes mellitus | 241 (25.1) | 6 (28.6) |
| Chronic obstructive pulmonary disease and/or asthma | 169 (17.6) | 5 (23.8) |
| Arthritis | 236 (24.6) | 3 (14.3) |
| Cancer excluding prostate cancer | 161 (16.8) | 6 (28.6) |
| Prostate cancer | 129 (13.5) | 6 (28.6) |
| Functional status (clinic-based screening tool) | | |
| Independent in ADLs | 889 (92.7) | 16 (76.2) |
| Difficulty with 1 or more ADLs | 70 (7.3) | 5 (23.8) |
| Help with 1–2 ADLs | 10 (1.0) | 1 (4.8) |
| Help with 3+ ADLs | 49 (5.1) | 2 (9.5) |
| Independent in IADLs | 821 (85.6) | 14 (66.7) |
| Difficulty with 1 or more IADLs | 138 (14.4) | 7 (33.3) |
| Help with 1–2 IADLs | 6 (0.6) | 0 (0.0) |
| Help with 3 or more IADLs | 129 (13.5) | 7 (33.3) |
| Hospitalization during the past year | 286 (29.8) | 6 (28.6) |

Abbreviations: ADL, activity of daily living; IADL, instrumental activity of daily living.

Percentages may not add to 100 due to rounding.

ADLs included bathing, dressing, transferring, toileting, and eating and IADLs included shopping, preparing food, managing medications, managing finances, doing housework, using transportation, and using the telephone.

## Educational session evaluation

LVNs and PCPs highly rated the overall quality of the educational session (median score 5, interquartile range [IQR], 1), with similar ratings for communication of project goals, clarity of learning objectives, and clarity of instruction (Table 3). LVNs and PCPs also highly rated the likelihood that they would apply what they learned to their work and their level of openness to adopting the new intervention (median score 4, IQR 1, for each measure). Retrospective knowledge scores before and after the intervention were high, with a median rating of 5

**Table 3. Evaluation of educational session by clinicians[a].**

| Questions | Median (IQR) | |
|---|---|---|
| *Please rate the quality of the following aspects of the presentation[b] (5-point scale from poor (1) to excellent (5))* | | |
| Communication of project goals | 5 (1) | |
| Clarity of learning objectives | 5 (1) | |
| Clarity of instruction for electronic screening tool | 5 (1) | |
| Overall quality of educational session | 5 (1) | |
| *Please rate the effectiveness of today's session[c] (5-point scale from very low (1) to very high (5))* | | |
| Likelihood you will apply what you learned to your practice/work | 4 (1) | |
| Your level of openness to adopting new approach | 4 (1) | |
| *Please rate your knowledge and ability before versus after today's session[d] (5-point scale from definitely cannot (1) to definitely can (5))* | | |
| Describe the importance of functional assessment for older adults in primary care | 5 (0) | 5 (0) |
| Measure functional status for older adults | 5 (1) | 5 (1) |
| Communicate with colleagues about functional status | 5 (1) | 5 (0) |

[a]Sixteen clinicians completed the evaluation.

[b]Rated on a 5-point Likert scale (5 excellent, 1 poor).

[c]Rated on a 5-point Likert scale (5 very high, 1 very low).

[d]Rated on a 5-point Likert scale (5 definitely can, 1 definitely cannot).

for self-rated ability to describe the importance of functional assessment for older adults in primary care, measure functional status in older adults, and communicate with colleagues about functional status.

In open-ended responses, LVNs and PCPs reported several plans to change their professional practice in response to the educational session. Planned changes included feeling "more motivated" to make time for functional status assessment and using the results of the screener to "refer patients to specific services," including geriatrics and rehabilitation.

## Implementation outcomes

**Appropriateness: Clinician perspectives.** In pre- and post-surveys, clinicians highly rated the intervention for its ability to meet patients' needs. The median rating was "very strong" (5, IQR 0) for three dimensions of appropriateness, including appropriateness based on needs and preferences of VA patients, participants' clinical experience, and consistency with clinical practices accepted by VA patients (Table 4).

Findings from qualitative interviews with LVNs and PCPs provided additional context for each dimension of appropriateness. First, LVNs viewed the new intervention as appropriate for addressing the needs of the aging population cared for at the VA. As an LVN noted,

"Our population. . .the majority of them are over 65, so when this [electronic tool] rolled out, I'm like 'yes!', because our population is getting older. We can see that they're declining physically, mentally,. . .psychosocial[ly], [and] this [tool] is more in-depth when we're screening them."

**Table 4. Pre- and post-implementation surveys completed by clinicians[a].**

| Implementation domain[b] and associated survey question(s) | Scores (Median, IQR) | |
|---|---|---|
| | Pre- | Post- |
| Acceptability, intervention characteristics, evidence strength and quality | | |
| In your opinion, please rate the strength of the evidence supporting the following statement: "Assessing functional status is necessary to improve care and outcomes for older patients"[c] | 4 (1) | 5 (1) |
| The functional assessment pilot study is supported by scientific evidence[d] | 5 (1) | 5 (1) |
| Appropriateness, outer setting, patient needs and resources | | |
| The functional assessment pilot study is appropriate based on your clinical experience with patients[b] | 5 (0) | 5 (0) |
| The functional assessment pilot study is consistent with clinical practices that have been accepted by VA patients[b] | 5 (0) | 5 (0) |
| The functional assessment pilot study takes into consideration the needs and preferences of VA patients[b] | 5 (0) | 5 (0) |
| Feasibility, inner setting, networks/communication | | |
| Staff members in your clinic cooperate to maintain and improve effectiveness of patient care[b] | 5 (0) | 5 (0) |
| Feasibility, inner setting, implementation climate | Survey, interview | |
| Staff members in your clinic are willing to innovate and/or experiment to improve clinical procedures[b] | 5 (0) | 5 (1) |
| Process, engaging champions | Survey | |
| This intervention has effective and knowledgeable clinic champions[b] | 5 (1.5) | 5 (1) |

[a]Sixteen clinicians completed the pre-implementation survey and 13 competed the post-implementation survey.

[b]Domains based on Consolidated Framework for Implementation Research (CFIR) constructs.

[c]Rated on a 5-point Likert scale (5 very strong evidence, 1 very weak evidence).

[d]Rated on a 5-point Likert scale (5 strongly agree, 1 strongly disagree).

Second, LVNs noted that measuring functional status is appropriate because without screening, impairments may go undetected. For example, LVNs reported that some patients look independent but actually need help with some tasks, and many patients won't bring up functional impairments on their own. As one LVN said, "Most patients won't ask for resources, but when you do the [screening] with them, it becomes apparent who needs help."

LVNs further noted that the screening tool was specific enough that it could point to how someone might need help in one area versus another (e.g., with bathing rather than transfer-ring). LVNs reported that compared to the previous screening tool, the new tool asked similar questions but was more in-depth, more specific, more tailored, and more effective at eliciting psychosocial factors that impact each patient. As one LVN stated, "I really think it's just more personalized, in a nutshell. It's more personalized and we can tailor it to that person, because John over there isn't gonna have the same points or same answers as Jane over here." In addition, LVNs noted that the new tool could encourage PCPs to address function during visits. As one LVN reflected, "we need...the doctors to be a little more proactive [about addressing function]."

Echoing LVNs, PCPs noted that making the screening tool mandatory forces PCPs to regu-larly discuss function, whereas otherwise they might only discuss it in the context of specific health events (e.g., a patient had a stroke). As one PCP shared,

"It might help me bring it up, whereas I might not have before. So, if it's the same patient I've been seeing for many years, and I don't notice anything new or different, I might not ask them are you having trouble with [ADLs]. But if a score that comes up from the LVN is high, then I might ask them, 'So, I see you've got some ADL needs. Is there anything causing you problems? Do you need some specific help with any of those? Do you feel like you've got the support that you need?'"

Like LVNs, PCPs noted that the screening tool helps patients by bringing up issues patients might otherwise not discuss. One PCP noted that the new tool is important "because I feel like sometimes patients don't think to bring up in a medical visit things that are more psychosocial-related, [like] 'I've been struggling to understand my finances.' A lot of times the way we find out about these things is if someone has a fall or a family member alerts us that they're struggling at home." Another PCP noted that identifying impairments was important because it allows clinicians to make appropriate referrals to meet patients' needs: "[A]t the VA, we are so lucky; we have so many services to help people, and when I do ultimately refer someone for PT or to [OT] and they get a shower chair or whatever it is that they need, they are typically so grateful and it improves their quality of life so much."

At the same time, PCPs noted that when there are competing clinical issues, it is not always helpful for the PCP to be forced into conversations about function. One PCP noted, "If the patient is there, say, for five problems, and then one of the [screening tools] is the ADL needs, that can eat up half of your appointment. So, it's just the practicality piece that's a challenge. It sort of forces you to go down that road and you just don't always have time for it."

**Appropriateness: Patient perspectives.** Patients generally reported that being asked about function is reasonable, legitimate, appropriate, and even essential to providing care. As one patient noted, "I think [the questions] are appropriate because there's a lot of people that need a hand." Another patient described how asking about function is particularly important for older adults, stating, "We appreciate that [the clinicians] are concerned, [because] especially as you get older, even under normal circumstances, it's difficult to maintain your activities of daily living, but even more so as you have health challenges."

'Like clinicians, patients reported that the screening tool could prompt patients to share concerns. One patient noted, "It's helpful that people ask you questions about how you are doing, because sometimes you don't think about those things or you don't think to mention them, whereas if the question is asked, you can respond to it." Several patients raised a contrary perspective, noting that assessing function was not appropriate because not all patients had impairments. As one patient said, "I am fairly robust and more like a 55-year-old than a 75-year-old. From my appearance it wouldn't make sense for them to ask that. It just wouldn't be appropriate." Another patient noted that they would independently bring up concerns related to function, noting, "Personally, I think I would probably bring it up before I was asked about it."

## Acceptability and satisfaction

In pre- and post-surveys, clinicians highly rated the intervention's acceptability in terms of the strength and quality of its supporting evidence (median pre- and post-scores 4 [IQR 1] and 5 [IQR 1]; median pre- and post-scores 5 [IQR 1], respectively; Table 4). On post-surveys, clinicians also highly rated their overall satisfaction (median score, 5 [IQR 1]). Patients highly rated their satisfaction with the screening tool for their experience with both the LVN screener and PCP assessment (5 [IQR, 0]).

In semi-structured interviews, clinicians generally perceived the new screening tool as acceptable to patients. LVNs noted that patients seemed to be open to being asked questions

about function and also appreciated the shorter format of the new tool. One LVN commented that patients will engage more with fewer questions, rather than with many, stating, "Since it's shorter, it's not too much wording. The patient listens to your questions more than when we were asking the older version, which was too many words and sometimes you can tell they don't want to answer." PCPs also noted that patients seem receptive to being asked about function and appreciative of being asked. As one PCP reflected, "I think patients are very appreciative of it and I really think it does enhance their perception of their physician as caring for them and caring about what's going on with them."

Patients similarly reported that the questions were acceptable and that they did not feel offended by being asked about function. As one patient shared, "There's a reason they're asking and I don't mind being asked." Instead, patients viewed the questions as reflecting their care teams' investment in their wellness. As one patient reflected, "All those questions are perfectly reasonable questions to ask. In fact, I'm glad to have them asked because it shows interest in my wellbeing." Patients further noted that screening for functional impairment could improve many aspects of patient care, including proactively identifying impairments, providing appropriate care, and informing preventive care approaches. As one patient commented, "Listen, if they ask personal questions, [then] they ask personal questions. . .if it gets down that they can take better care of me by asking me the questions, then by all means ask me the questions."

**Acceptability of screening tool design.**   Clinicians shared varying perspectives on the acceptability of the design of the screening tool. LVNs generally viewed the tool as an improvement on the previous tool, citing several features including the ability to add details in the optional free-text sections. As one LVN described,

> "I like this [screening tool] better than the other because you can add more details, rather than just having to fill in the bubbles. . . 'Cause a lot of the veterans, when they do have difficulty or get a positive assessment then we can go more into depth with the difficulties that they're struggling with, versus just bubbling in that they're having difficulty."

LVNs also commented that the wording of the new screening tool seemed less invasive and more sensitive to veteran concerns about being asked about functional decline compared to the previous tool. PCPs noted that the wording of the new reminder was more "conversational" and that the flow was more "practical."

One aspect of the new tool that raised concern from clinicians was the numeric scoring method. LVNs noted some confusion about how the scoring works and what the scores mean. Several PCPs also questioned the utility of a numeric score in assessing function, rather than examining the specific difficulties reported by a patient. In contrast, one PCP reported that some PCPs might prefer score-based reminders, because free text can be laborious to enter or review and varies greatly in quality depending on who enters it, contributing to "decision fatigue." This PCP noted that "Required free text sections are what contribute the most to decision fatigue, for me at least. . .For me, the free-text sections, the more that can be optional the better. . .although having them there as an option if you feel the need to elaborate is helpful."

PCP perspectives varied regarding the acceptability of the referral menu. Several PCPs liked the menu, with one PCP stating that it provides a summary of options to consider, and another noting that placing referrals earlier to support patients could help reduce clinical workload long-term. However, one PCP preferred leaving the PCP portion more open-ended so that they could make their own decision about next steps for clinical management.

Patients did not recommend specific changes to the screening tool and generally welcomed the addition of any questions with the potential to improve care.

## Feasibility

In pre- and post-surveys, clinicians highly rated the feasibility of implementing the intervention. Clinicians "strongly agreed" that staff members in their clinic cooperated to maintain and improve the effectiveness of patient care (median score 5, IQR 0 both pre- and post-) and that the intervention had effective and knowledgeable clinic champions (pre-survey median score 5, IQR 1.5; post-survey median score 5, IQR 1; Table 4).

In qualitative interviews, LVNs noted that the new screening tool was often shorter and more feasible to administer than the old tool, reducing time burden for both LVNs and patients and improving workflow. As one LVN noted, "It's easier for the workflow because if you get a negative you don't need to go too much into detail. . .versus if they get a positive score then [you] need to elaborate more, ask more questions."

Similarly, PCPs noted that the new screening tool was more streamlined for LVNs and less burdensome for PCPs than the previous tool. As one PCP noted, "I appreciate the simplicity up front." PCPs generally liked the distinct LVN and PCP portions of the screening tool because it made assessment more efficient and prompted the PCP to act when needed. One PCP commented, "I think dividing it into two parts has worked well for me personally by allowing the LVN to complete a portion of the reminder. It has been no additional work for me to take on that screening [tool]." Another PCP went on to explain how the reminder "meets a need of doing that work for you and letting you focus on reacting rather than taking the time to [screen], which, for some things that are relatively algorithmic and simple, is very helpful to take that off the list of things to do."

In terms of impact on clinic processes and workflows, LVNs described generally seamless adoption of the tool. PCPs similarly reported that the new tool did not interrupt existing workflows and felt "very invisible" from their perspective. Patients also shared that the questions on the new screening tool weren't very memorable and, if noticed at all, seemed to seamlessly blend into clinic processes.

**Feasibility of use during the COVID pandemic.** LVNs noted that clinical interactions during the pandemic became more formalized due to the shift to telehealth, limiting their use of informal cues to glean information on functional status (e.g., a patient struggling to walk to the bathroom). They also described the loss of some opportunities for care coordination that were commonplace before the pandemic, such as directly handing off a patient to an on-site social worker. One LVN explained, "Right now we're not really seeing that many patients [in person]–it's not like they can just come in and have a social worker just be right there with them to assist them or a provider who can assist them right then and there. [I]t's not like before [when a patient] would screen positive [and] we can diagnose the problem right away or have them show us how hard of a time they have walking."

Despite these changes, LVNs generally viewed the screening tool as being adaptable for use either in-person or via telehealth and retaining its usefulness during the pandemic. As one LVN reflected, "This [screening tool] is accessible, even in the midst of COVID." LVNs also reported that electronic team communication about patient functional status largely remained the same during the pandemic, as LVNs still used patient notes in the EHR to notify the PCP or social worker if the patient needed assistance with daily activities.

## Adoption

During the study period, all 959 eligible patients were screened for functional impairment (100% reach), with 100% of participating LVNs completing screening for one or more patients. In chart reviews for a subset of 50 patients with functional impairment, 48% had referrals ordered to address impairments (e.g., physical therapy, occupational therapy, prosthetics)

within 1 week of the visit when the functional status questionnaire was administered and 64% within 1 month, with 57% of PCPs placing referrals for one or more patients within 1 week and 77% of PCPs within 1 month. Seventy-eight percent of clinician notes associated with the visit when screening was completed had content related to functional status.

## Fidelity

In the subset of 21 patients who completed interviews with researchers, we examined the sensitivities, specificities, positive and negative predictive values, and positive and negative likelihood ratios of the two-part clinic-based screening tool for detecting functional impairment compared to the interview eliciting difficulty and need for help with all ADLs and IADLs (Figs 2 and 3 and Table 5). The sensitivity of the clinic-based tool was 84% (95% CI, 68–100) for detecting need for help with 1 or more ADLs with a specificity of 100% (95% CI, 100–100), while the sensitivity for detecting difficulty with 1 or more ADLs was 67% (95% 45–88) with a sensitivity of 100% (95% CI, 100–100). For detecting need for help with 1 or more IADLs, the sensitivity was 71% (95% CI, 49–92) with a specificity of 75% (95% CI, 33–100), while for difficulty with 1 or more IADLs the sensitivity was 65% (95% CI, 42–87) with a specificity of 100% (95% CI, 100–100). The kappa statistics for agreement between the clinic- versus research-collected data ranged from .33-.50 (Table 5).

## Sustainability

In qualitative interviews, clinicians provided several recommendations for how to improve the intervention and its implementation and sustain its use over time. LVNs noted that after completing the screening tool, they seldom see how it is used to inform patient care, which

| A | | Reference standard | | |
|---|---|---|---|---|
| | | Difficulty in 1 or more ADLs | Independent in all ADLs | **Total** |
| **VA functional status data** | Difficulty in 1 or more ADLs | 3 | 0 | **3** |
| | Independent in all ADLs | 6 | 12 | **18** |
| | **Total** | **9** | **12** | **21** |

| B | | Reference standard | | |
|---|---|---|---|---|
| | | Needs help in 1 or more ADLs | Independent in all ADLs | **Total** |
| **VA functional status data** | Needs help in 1 or more ADLs | 2 | 0 | **2** |
| | Independent in all ADLs | 3 | 16 | **19** |
| | **Total** | **5** | **16** | **21** |

**Fig 2. Agreement of VA functional status data with reference standard of research-collected data for assessing ability to perform activities of daily living.** (A) Agreement for difficulty with 1 or more activities of daily living. (B) Agreement for needing help with 1 or more activities of daily living.

| A | | Reference standard | | |
|---|---|---|---|---|
| | | Difficulty in 1 or more IADLs | Independent in all IADLs | **Total** |
| **VA functional status data** | Difficulty in 1 or more IADLs | 4 | 0 | **4** |
| | Independent in all IADLs | 6 | 11 | **17** |
| | **Total** | **10** | **11** | **21** |

| B | | Reference standard | | |
|---|---|---|---|---|
| | | Needs help in 1 or more IADLs | Independent in all IADLs | **Total** |
| **VA functional status data** | Needs help in 1 or more IADLs | 3 | 1 | **4** |
| | Independent in all IADLs | 5 | 12 | **17** |
| | **Total** | **8** | **13** | **21** |

**Fig 3. Agreement of VA functional status data with reference standard of research-collected data for assessing ability to perform instrumental activities of daily living.** (A) Agreement for difficulty with 1 or more instrumental activities of daily living. (B) Agreement for needing help with 1 or more instrumental activities of daily living.

contributed to LVNs feeling less integral to the clinical team. As one LVN noted, "I don't know what the doctors do after [I complete the screening tool]. . . I guess I just hope that the patient is getting their needs met. . .Yeah, I don't know what the doctor is doing on their end."

Both LVNs and PCPs recommended that LVNs take on a more active role in using the functional status data, by giving LVNs the ability to order equipment and referrals based on the results of the screening tool (e.g., bedside commode, physical therapy consult), which the PCP would then review and sign. As one PCP recommended, "It would be something to ask LVNs if they would like to be able to [complete] certain consults. I think sometimes those things can be empowering. . . [it] might make them feel like they are contributing a little more."

## Discussion

In this study, we pilot-tested a person-centered interprofessional intervention to improve identification and management of functional impairment among older adults in VA primary

**Table 5. Test characteristics of VA functional status data compared with the reference standard of research-collected data.**

| | Sensitivity (95% CI) | Specificity (95% CI) | Positive Predictive Value (95% CI) | Negative Predictive Value (95% CI) | Positive Likelihood Ratio (95% CI) | Negative Likelihood Ratio (95% CI) | Kappa statistic |
|---|---|---|---|---|---|---|---|
| Difficulty with 1 or more ADLs[a] | 67 (45, 88) | 100 (100, 100) | 100 (100, 100) | 33 (25, 64) | Undefined | .33 (.12, .55) | 0.36 (0.03, 0.70) |
| Needs help with 1 or more ADLs | 84 (68, 100) | 100 (100, 100) | 100 (100, 100) | 40 (0, 83) | Undefined | .16 (.00, .32) | 0.50 (0.05, 0.96) |
| Difficulty with 1 or more IADLs | 65 (42, 87) | 100 (100, 100) | 100 (100, 100) | 40 (10, 70) | Undefined | .35 (.13, .58) | 0.41 (0.09, 0.73) |
| Needs help with 1 or more IADLs | 71 (49, 92) | 75 (33, 100) | 92 (78, 100) | 38 (4, 71) | 2.82 (.73, Undefined) | .39 (.08, 1.57) | 0.33 (-0.06, 0.72) |

care settings. Clinicians and patients reported that the intervention was appropriate, accept-able, and feasible to complete, even during the COVID pandemic. Adoption and reach were high, with 100% of LVNs completing screening for functional impairment, 100% of eligible patients receiving screening, and 48% of patients with identified impairment receiving refer-rals to address impairments within 1 week. The accuracy of the two-step screening approach for detecting functional impairment ranged from fair to very good, suggesting opportunities to improve sensitivity while limiting clinician burden. Taken together, these findings suggest that this intervention is a promising approach to improve identification and management of func-tional impairment among older patients in primary care.

To our knowledge, few recent interventions have been developed and implemented that specifically focus on increasing routine, standardized measurement of functional status for older adults in primary care settings [44]. However, a range of interventions to help improve functional status and prevent functional decline depend on understanding function. For exam-ple, patients with identified functional impairment benefit from referrals to physical and occu-pational therapy [9]. In addition, multi-component interventions have been shown to benefit patients with functional impairment, including Community Aging in Place–Advancing Better Living for Elders (CAPABLE) and Geriatric Resources for Assessment and Care of Elders (GRACE) [10,45–47]. Identification of functional impairments also informs delivery of per-son-centered care, including individualizing cancer screening and referring patients for needed services and supports [16]. This study provides an approach to widely implement screening for functional impairment, allowing for systematic identification of older adults who may benefit from referral to interventions and allowing for person-centered care that accounts for functional status. This study also provides an approach to identify earlier stages of func-tional impairment (i.e., when participants have difficulty performing daily activities but do not yet need help) [48]. Earlier identification could allow for earlier intervention, with the potential to help prevent or delay subsequent functional decline and associated health care utilization.

Our findings also suggest that it is possible to address key barriers to functional status screening in primary care, including time limitations and burdensome electronic tools. Clini-cians perceived the intervention as appropriate, acceptable, and feasible to implement, noting that the brief pre-screener portion of the electronic tool reduced time burden, making it more feasible to measure functional status. Clinicians also perceived the design of the tool as flexible enough to fit in existing workflows. Consistent with the high acceptability of the intervention, adoption by LVNs and PCPs was high, with all participating LVNs completing screening for functional impairment and nearly half of patients with identified impairments receiving refer-rals within 1 week of screening.

Our findings have implications for how to improve and sustain measurement of functional status measurement in primary care settings. The sensitivity and specificity of the two-step screening approach for detecting ADL and IADL impairment compared to full screening ran-ged from fair to very good, although confidence intervals were wide given the small sample size. These findings provide support for the overall accuracy of the clinic-collected measures while also suggesting how to improve sensitivity for detecting functional impairment. Several partici-pants who screened negative on the ADL pre-screener in clinic screened positive for difficulty walking on the full screener administered by the research team. Similarly, several participants who screened negative on the IADL pre-screener in clinic screened positive for difficulty man-aging medications on the full screener. These findings suggest that modifying the ADL and IADL pre-screener to include the ADLs and IADLs that were most frequently missed has the potential to capture individuals who screened out but had impairments. Second, due to the design of the EHR, it was not possible to automatically calculate a numeric functional status score within the electronic screening tool, posing a burden for LVNs. Automatically calculating

functional status scores in the EHR could address this barrier while providing population-level data to understand the level of functional impairment among patients cared for in primary care practices, medical centers, and regions. Third, prior studies have shown that a key barrier to implementing functional status measurement is the perception that the data collected will not be used to improve care [22]. Nurses in this study voiced a similar concern, noting that they were unaware of how the data they collected were used by PCPs. Both nurses and PCPs reflected that granting front-line nurses the ability to enter orders that could be reviewed and signed by PCPs–such as physical therapy, occupational therapy, or prosthetics–could empower nurses to impact patient care while reducing clinical burden for PCPs. While not all clinical settings allow nurses to enter orders for PCP review and signature, this approach has the potential to improve management of functional impairment while increasing interdisciplinary collaboration and improving nursing satisfaction and morale [49,50].

This study has several limitations. Due to the competing demands posed by the COVID pandemic, we were unable to create clinical reports on functional status for clinicians and health system leaders. Thus, we could not evaluate outcomes associated with this intervention component. Because this was a pilot study with a pre-to-post design, there was no control group for comparison. We evaluated process measures reflecting assessment of functional impairment, e.g., referrals, rather than measures of functional status or health care utilization over time; larger scale studies are needed to test these outcomes. We tested the intervention at two VA medical centers in Northern California, both of which previously had an electronic functional status screening tool in use. Our findings may not be generalizable to other geographic areas or to clinics without prior experience with electronic functional status screening. The VA differs from non-VA clinical settings in several aspects, including longer appointment times, a predominantly male patient population [51], and widespread implementation of the patient-centered medical home model in primary care [52]. However, barriers and facilitators to functional status measurement–which this intervention was designed to address and leverage–are similar across VA and non-VA primary care settings [22,53–57]. We used an age cut-off of 75 and older for the intervention in order to align with existing VA screening measures among older adults. Because the prevalence of functional impairment increases with age, this may have resulted in a higher prevalence of functional impairment than in a study using a younger age cut-off.

If these findings are replicated in larger studies, this intervention may have important implications for clinical care, public health, and research. Proactively measuring functional status has the potential to improve identification and management of functional status for older adults on a broad scale, leading to improvements in population health, consistent with VA's goal of becoming the largest Age-Friendly Health System in the United States [58,59]. For clinicians and researchers, systematically measuring functional status in primary care settings would also provide a platform to implement and test the effectiveness of interventions to improve functional status among older patients.

## Conclusions

We developed a novel person-centered interprofessional intervention to improve identification and management of functional impairment among older patients in VA primary care settings. Clinicians and patients reported that the intervention was appropriate, acceptable, and feasible to complete, even during the COVID pandemic, with high adoption and fair to very good fidelity. These findings suggest that this intervention is a promising approach to improve identification and management of functional impairment among older patients in primary

care. Further refinement and broader implementation and evaluation of this intervention is currently underway in VA primary care settings.

## Acknowledgments

The authors do not have any additional contributors to report.

## Author Contributions

**Conceptualization:** Rebecca T. Brown, Kara Zamora, Anael Rizzo, Malena J. Spar, Francesca M. Nicosia.

**Data curation:** Rebecca T. Brown, Kara Zamora, Anael Rizzo, Malena J. Spar, Francesca M. Nicosia.

**Formal analysis:** Rebecca T. Brown, Kara Zamora, Anael Rizzo, Malena J. Spar, Kathy Z. Fung, Francesca M. Nicosia.

**Funding acquisition:** Rebecca T. Brown.

**Investigation:** Rebecca T. Brown, Kara Zamora, Anael Rizzo, Malena J. Spar, Kathy Z. Fung, Lea Santiago, Annie Campbell, Francesca M. Nicosia.

**Methodology:** Rebecca T. Brown, Kara Zamora, Anael Rizzo, Malena J. Spar, Kathy Z. Fung, Francesca M. Nicosia.

**Project administration:** Rebecca T. Brown, Francesca M. Nicosia.

**Resources:** Rebecca T. Brown, Lea Santiago, Annie Campbell.

**Supervision:** Rebecca T. Brown, Francesca M. Nicosia.

**Validation:** Rebecca T. Brown, Kara Zamora, Anael Rizzo, Malena J. Spar, Kathy Z. Fung, Francesca M. Nicosia.

**Writing – original draft:** Rebecca T. Brown, Kara Zamora, Francesca M. Nicosia.

**Writing – review & editing:** Rebecca T. Brown, Kara Zamora, Anael Rizzo, Malena J. Spar, Kathy Z. Fung, Lea Santiago, Annie Campbell, Francesca M. Nicosia.

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
