## [Decision Letter · Decision Letter 0]

9 Jan 2024

PONE-D-23-34956Improving measurement of functional status among older adults in primary care: a pilot studyPLOS ONE

Dear Dr. Brown,

Thank you for submitting your manuscript to PLOS ONE. After careful consideration, we feel that it has merit but does not fully meet PLOS ONE’s publication criteria as it currently stands. Therefore, we invite you to submit a revised version of the manuscript that addresses the points raised during the review process.

 Please submit your revised manuscript by Feb 23 2024 11:59PM. If you will need more time than this to complete your revisions, please reply to this message or contact the journal office at plosone@plos.org. Please include the following items when submitting your revised manuscript:A rebuttal letter that responds to each point raised by the academic editor and reviewer(s). You should upload this letter as a separate file labeled 'Response to Reviewers'.A marked-up copy of your manuscript that highlights changes made to the original version. You should upload this as a separate file labeled 'Revised Manuscript with Track Changes'.An unmarked version of your revised paper without tracked changes. You should upload this as a separate file labeled 'Manuscript'.If applicable, we recommend that you deposit your laboratory protocols in protocols.io to enhance the reproducibility of your results. Protocols.io assigns your protocol its own identifier (DOI) so that it can be cited independently in the future. For instructions see: https://journals.plos.org/plosone/s/submission-guidelines#loc-laboratory-protocols. Additionally, PLOS ONE offers an option for publishing peer-reviewed Lab Protocol articles, which describe protocols hosted on protocols.io. Read more information on sharing protocols at https://plos.org/protocols?utm_medium=editorial-email&utm_source=authorletters&utm_campaign=protocols.

We look forward to receiving your revised manuscript.

Kind regards,

De-Chih Lee, Ph.D.

Academic Editor

PLOS ONE

Journal Requirements:

3. Please describe in your methods section how capacity to provide consent was determined for the participants in this study. Please also state whether your ethics committee or IRB approved this consent procedure. If you did not assess capacity to consent please briefly outline why this was not necessary in this case.

4. In the ethics statement in the Methods, you have specified that verbal consent was obtained. Please provide additional details regarding how this consent was documented and witnessed, and state whether this was approved by the IRB.

5. In the online submission form, you indicated that "The underlying data for this study consists of surveys and in-depth, qualitative interviews with (1) older Veterans and (2) employees of the U.S. Veterans Health Administration. It is not possible to create a minimal data set with this survey and qualitative data as this study did not obtain ethical approval or informed consent from participants to publicly share underlying survey and qualitative data sets. Relevant excerpts from transcripts are included within the paper. The datasets generated and/or analyzed during this study are not publicly available but may be available upon request at the Center for Health Equity Research and Promotion of the U.S. Department of Veterans Affairs administrative offices, at 215-823-5817."

Additional Editor Comments:

Please make minor revisions according to the comments of the two reviewers.

Reviewers' comments:

Reviewer's Responses to Questions

**Comments to the Author**

1. Is the manuscript technically sound, and do the data support the conclusions?

Reviewer #1: Yes

Reviewer #2: Yes

2. Has the statistical analysis been performed appropriately and rigorously? 

Reviewer #1: Yes

Reviewer #2: Yes

3. Have the authors made all data underlying the findings in their manuscript fully available?

Reviewer #1: Yes

Reviewer #2: No

4. Is the manuscript presented in an intelligible fashion and written in standard English?

Reviewer #1: Yes

Reviewer #2: Yes

5. Review Comments to the Author

Reviewer #1: Comments PLOS One

Dear Editor,

This article deals with a very interesting topic. The relevance of this article is undeniable. Additionally, the article is well structured, the methods of analysis are relevant, and the tables are informative. However, before its publication, some changes need to be made.

You will find my comments below.

Sincerely yours

In the fidelity section on page 32, you should be careful with confidence intervals given your sample (n=21). I therefore suggest that the authors give the limits of these results.

Reviewer #2: Thank you for the opportunity to review this excellent manuscript on improving measurement of functional status among older adults in primary care. This is a well written paper on an important topic that will be of great interest to readers seeking way to improve care of older adults. Please consider the following suggestions which may strengthen this manuscript:

- Introduction - consider using the term 'person centered' instead of 'patient centered' (see here https://www.ncbi.nlm.nih.gov/pmc/articles/PMC6069658/)

- consider adding understanding frailty after 'assessing prognosis'

- page 5, paragraph two - consider giving an example of routine outpatient screenings in the VA (do you mean things like PHQ2 for depression?)

- page 9 - para 2 - was it any kind of clinica appointment or only routine PCP (vs urgent visit)?

- page 13 - would specific on the last line that it was pandemic related restrictions that limited ethnographic observations

Methods - this is so clearly delineated and easy for the reader to follow, thank you!

Would describe a bit about what the educational session entailed (maybe as a supplemental table that outlines the session?)

Results

Table 2 - in the legend would list what you included in your definition of ADLs (Bathing etc) and iADLS

page 22 - can you give an example of the types of practice changes participants reported intending to try?

The results section is easy to follow and clearly reported.

Discussion:

- consider explaining why you picked age 75 and older for the intervention and how that might affect prevalence of functional impairment in your pilot

- page 35 last line - again consider 'person-centered' instead of patient centered

- page 36, first paragraph - what do you mean by "just need for help"?

- page 36 - second last para, the adoption and reach repeats nearly verbatim what's in the opening of the discussion so would rephrase

- consider mentioning here or in the intro the importance of this work to the VA's goal of becoming an Age-Friendly health system (could cite https://agsjournals.onlinelibrary.wiley.com/doi/10.1111/jgs.18070 and/or https://www.ncbi.nlm.nih.gov/pmc/articles/PMC10204932/

This is such a well written article on such an important topic - thank you for the incredible work you are doing!

6. PLOS authors have the option to publish the peer review history of their article (what does this mean?). If published, this will include your full peer review and any attached files.

Reviewer #1: No

Reviewer #2: No

---

## [Author Response · Author response to Decision Letter 0]

13 Feb 2024

Please see attached Response Letter.

---

## [Decision Letter · Decision Letter 1]

5 Mar 2024

PONE-D-23-34956R1Improving measurement of functional status among older adults in primary care: a pilot studyPLOS ONE

Dear Dr. Brown,

Thank you for submitting your manuscript to PLOS ONE. After careful consideration, we feel that it has merit but does not fully meet PLOS ONE’s publication criteria as it currently stands. Therefore, we invite you to submit a revised version of the manuscript that addresses the points raised during the review process.

We look forward to receiving your revised manuscript.

Kind regards,

De-Chih Lee, Ph.D.

Academic Editor

PLOS ONE

Journal Requirements:

Additional Editor Comments:

Some references are out of date. Unless it's an important theory, please find more recent references to replace these older ones if possible.

Reviewers' comments:

Reviewer's Responses to Questions

**Comments to the Author**

1. If the authors have adequately addressed your comments raised in a previous round of review and you feel that this manuscript is now acceptable for publication, you may indicate that here to bypass the “Comments to the Author” section, enter your conflict of interest statement in the “Confidential to Editor” section, and submit your "Accept" recommendation.

Reviewer #1: All comments have been addressed

Reviewer #2: All comments have been addressed

2. Is the manuscript technically sound, and do the data support the conclusions?

Reviewer #1: Yes

Reviewer #2: Yes

3. Has the statistical analysis been performed appropriately and rigorously? 

Reviewer #1: Yes

Reviewer #2: Yes

4. Have the authors made all data underlying the findings in their manuscript fully available?

Reviewer #1: Yes

Reviewer #2: No

5. Is the manuscript presented in an intelligible fashion and written in standard English?

Reviewer #1: Yes

Reviewer #2: Yes

6. Review Comments to the Author

Reviewer #1: Thank you for giving me the opportunity to revise this new version of the article. I am satisfied with the corrections made by the authors. The authors have taken into account my recommendations and suggestions. Consequently, the article can be published as is.

Reviewer #2: Thank you for thoroughly addressing all the reviewer comments. This is an excellently done manuscript and I enjoyed the opportunity to review it - congratulations on this important work!

7. PLOS authors have the option to publish the peer review history of their article (what does this mean?). If published, this will include your full peer review and any attached files.

Reviewer #1: No

Reviewer #2: No

---

## [Author Response · Author response to Decision Letter 1]

27 Mar 2024

March 6, 2024

De-Chih Lee, Ph.D.

Academic Editor

PLOS ONE

Dear Dr. Lee,

We thank the Editor for their helpful comments and appreciate the opportunity to address the comments and revise our manuscript. Below, please find item-by-item responses to the comments, which are included verbatim. All page and paragraph numbers refer to locations in the revised manuscript. We have uploaded a clean copy of the revised manuscript as well as a copy of the manuscript with tracked changes.

Responses to Journal Requirements:

1. Comment: Please review your reference list to ensure that it is complete and correct. If you have cited papers that have been retracted, please include the rationale for doing so in the manuscript text, or remove these references and replace them with relevant current references. Any changes to the reference list should be mentioned in the rebuttal letter that accompanies your revised manuscript. If you need to cite a retracted article, indicate the article’s retracted status in the References list and also include a citation and full reference for the retraction notice.

a. Response: We have reviewed our reference list to ensure that it is complete and correct and have updated multiple references as discussed below. We have ensured that the list does not include retracted papers.

Responses to Editor:

1. Comment: Some references are out of date. Unless it's an important theory, please find more recent references to replace these older ones if possible.

a. Response: We have extensively updated the reference list to remove older references and provide newer ones. For some seminal references that helped to define the field and for those related to theory, we have retained older references. For example, we retain key references from Fried (NEJM, 2002) and Walter (JAMA, 2001). Please note that Microsoft Word does not track changes to references made using Endnote software.

We thank the Editor for their time and efforts to improve our manuscript. 

Sincerely,

Rebecca Brown

---

## [Editor Report · Decision Letter 2]

24 Apr 2024

Improving measurement of functional status among older adults in primary care: a pilot study

PONE-D-23-34956R2

Dear Dr. Brown,

We’re pleased to inform you that your manuscript has been judged scientifically suitable for publication and will be formally accepted for publication once it meets all outstanding technical requirements.

Kind regards,

De-Chih Lee, Ph.D.

Academic Editor

PLOS ONE

Additional Editor Comments (optional):

All comments have been addressed.
---

## [Editor Report · Acceptance letter]

30 Apr 2024

PONE-D-23-34956R2 

PLOS ONE

Dear Dr. Brown, 

I'm pleased to inform you that your manuscript has been deemed suitable for publication in PLOS ONE. Congratulations! Your manuscript is now being handed over to our production team.

Kind regards, 

on behalf of

Dr. De-Chih Lee 

Academic Editor

PLOS ONE